# Targeting MAPK Signaling in Cancer: Mechanisms of Drug Resistance and Sensitivity

**DOI:** 10.3390/ijms21031102

**Published:** 2020-02-07

**Authors:** Shannon Lee, Jens Rauch, Walter Kolch

**Affiliations:** 1Systems Biology Ireland, School of Medicine, University College Dublin, Dublin 4, Ireland; shannon.lee1@ucdconnect.ie (S.L.); jens.rauch@ucd.ie (J.R.); 2School of Biomolecular and Biomedical Science, University College Dublin, Dublin 4, Ireland; 3Conway Institute of Biomolecular & Biomedical Research, University College Dublin, Dublin 4, Ireland

**Keywords:** MAPK, ERK, JNK, p38, cancer, drug resistance, combination therapy, metabolism, epigenetics

## Abstract

Mitogen-activated protein kinase (MAPK) pathways represent ubiquitous signal transduction pathways that regulate all aspects of life and are frequently altered in disease. Here, we focus on the role of MAPK pathways in modulating drug sensitivity and resistance in cancer. We briefly discuss new findings in the extracellular signaling-regulated kinase (ERK) pathway, but mainly focus on the mechanisms how stress activated MAPK pathways, such as p38 MAPK and the Jun N-terminal kinases (JNK), impact the response of cancer cells to chemotherapies and targeted therapies. In this context, we also discuss the role of metabolic and epigenetic aberrations and new therapeutic opportunities arising from these changes.

## 1. Introduction

MAPK pathways are cascades of three kinases, where the most upstream kinase (MAPKKK) responds to various extra- and intracellular signals and activates the middle kinase (MAPKK) by direct phosphorylation. MAPKKs exclusively phosphorylate and activate a MAPK, which typically has many substrates that execute specific cell fate decisions adequate to the input signal [1] (Figure 1).

MAPK substrate phosphorylation often includes the inhibition of upstream activators. This configuration corresponds to a negative feedback amplifier that combines signal amplification through the 3-tiered kinase cascade with a negative feedback from the output back to the input signal, thereby ensuring robustness against noise and graded responses [2]. MAPKs react to a wide variety of input signals including physiological cues such as hormones, cytokines, and growth factors, as well as endogenous stress and environmental signals. Thus, they are traditionally classified in mitogen and stress activated MAPKs, with classic representatives being ERK as mitogen responsive and JNK and p38 as stress responsive MAPKs. Physiologically, the distinction is blurry with all three families responding to a wide and overlapping variety of signals. MAPK signaling is altered in many diseases [3] and its kinase components have, therefore, been in the crosshairs of drug development for the last two decades.

The farthest progress has been made in cancer and with drugs targeting the RAS-RAF-MEK-ERK pathway. Prolific work has been done on drugs targeting this pathway and elucidating mechanisms of sensitivity and resistance. As the results have been extensively reviewed [4,5,6,7,8,9,10,11,12,13], we only briefly summarize the salient findings here. Instead, we focus on discussing less well reviewed areas of MAPK signaling and their relevance to drug resistance, i.e., the JNK and p38 MAPK pathways, as well as epigenetic and metabolic changes linked to MAPK signaling.

## 2. Mechanisms of Drug Resistance in the ERK Pathway

The RAS-RAF-MEK-ERK pathway is altered in ~40% of all human cancers, mainly due to mutations in BRAF (~10%) and its upstream activator RAS (~30%) [14]. MEK inhibitors were the first drugs developed, but despite their high potency and selectivity largely disappointed in the clinic [4,15]. This failure is attributable to the negative feedback amplifier property of the pathway, which autocorrects perturbations to the amplifier, i.e., MEK, to keep ERK signaling intact [2]. That means unless the amplifier kinase MEK is inhibited almost completely, there is little effect on the output strength, i.e., ERK activation (Figure 2). This work also predicted that breaking the negative feedback loop by inhibiting its target RAF will allow MEK inhibitors to work. Indeed, the combination of RAF and MEK inhibitors is now standard in the therapy of metastatic malignant melanoma and other cancer types [5,6,7,8,9,10].

Most of the seminal work was done in metastatic malignant melanoma, which is hallmarked by a high prevalence of BRAF (50–60%) and NRAS (15–20%) mutations [14]. RAF and MEK inhibitors are effective in BRAF mutated but not NRAS mutated melanomas (see below). Despite very high initial response rates, relapse is frequent, and a whirlwind of research work has discovered a plethora of resistance mechanisms. Classically, drug resistance was considered to be caused by mutations in the target protein that interfere with drug binding, elimination of the drug from the target cell by transporters, or enhanced degradation [16]. Resistance to RAF and MEK inhibitors brought a new mechanism into the limelight. There, the prevalent escape routes are adaptive network responses that result in the revival of ERK activation or the recruitment of other pathways that can substitute for ERK activity (Figure 3).

Although a main mechanism of resistance to RAF or MEK inhibitors is reactivation of ERK signaling [18,19,20], resistance mechanisms to combined RAF and MEK inhibition increasingly include the activation of alternative pathways that can drive cancer cell proliferation and survival [21,22]. The resistance mechanisms to clinically used inhibitors that block ERK signaling are summarized in Table 1.

Network adaptations as a source of drug resistance are now emerging as an important mechanism in other therapeutics areas [45,46], and can even contribute to intrinsic drug resistance arising from stochastic network fluctuations that convey resistance [29]. This is an important insight with immediate impact on how we design drug therapies. We have to replace the current view of breaking oncogenic signaling by hitting a driver molecule, e.g., a mutated oncogene, hard enough by a view that considers network context. Network adaptations employ a variety of intricate circuitries, such as cross-talk, feedback, and feedforward loops that make it often difficult to anticipate or even trace the decisive adaptations that confer drug resistance. Thus, computational and mathematical models that allow simulating signaling networks and predicting adaptive responses will become essential tools to get a handle on the plasticity of signaling networks and outwit their adaptive capacity [47,48,49,50].

In the case of metastatic malignant melanoma, the restoration of ERK signaling appears to be of major clinical relevance, and there the most important mechanism hinges on RAF dimerization. The formation of dimers is part of the physiological RAF activation mechanism [51] and is initiated by binding of RAF to RAS. Thus, RAF kinases are constitutively dimerized in tumors with mutant RAS [28,52]. Furthermore, RAF inhibitors can promote dimerization, and alternative splicing of mutant BRAF can also give rise to constitutive dimers [27]. Interestingly, dimers are drug resistant and actually can cause “paradoxical” activation of the ERK pathway as one drug bound protomer can activate its drug free binding partner through allosteric transactivation [28,52]. It is counterintuitive that a drug in large stoichiometric excess of RAF should only be able to bind to one RAF molecule in a dimer, but it was experimentally observed [53] and is explainable on thermodynamic grounds [54], which include a slight structural asymmetry of RAF protomers in the dimer [55] (Figure 4). This type of resistance can be broken by RAF inhibitors that prevent dimerization [56], or by a combination of two structurally different RAF inhibitors that target the two different conformations of RAF proteins in a dimer. For the latter approach, Kholodenko and colleagues developed a new mathematical model that considers the genetic mutational background, network context, thermodynamic parameters, and post-translational modifications and can accurately predict which type of inhibitors should be combined [57].

Importantly, these strategies also can block mutant RAS signaling. These developments highlight that for the development of efficacious drugs intricacies at the single molecule and network level must be understood and integrated.

## 3. The Role of the JNK Pathway in Drug Resistance

The JNK family is encoded by three genes: JNK1, JNK2, and JNK3. JNK1 and JNK2 are ubiquitously expressed, whereas JNK3 expression is tissue specific with highest levels in the brain, heart and testis [58]. JNK was discovered as kinase activity that relays oncogenic RAS transformation to nuclear transcription by phosphorylating and activating JUN transcription factors [59]. When the kinase was cloned, it turned out that it also is activated by stress signals, such as ultraviolet irradiation [60]. This finding gave rise to the notion that oncogenic and stress signaling may be intertwined. Further research tried to disentangle this relationship, leading to the view that ERK mainly stimulates proliferation in response to mitogenic growth factors, whereas JNK mainly promotes cell stress responses and apoptosis triggered by inflammatory cytokines and environmental insults. Early results that these pathways often act as antagonists and mutually inhibit each other [61] confirmed this distinction. However, a host of research over the last 25 years has nuanced this simple concept. We now have reverted to the original observation that JNK has dual functions and can mediate both cell transformation and apoptosis through a variety of mechanisms that partially overlap with ERK [62,63]. Genetic evidence also supports this dual role. JNK1/2 can suppress tumorigenesis in the skin, intestine, breast, prostate, and hematological system [64,65]. For instance, in breast cancer, the upstream JNK activator MKK4 is often mutated, and deletion of JNK1 and JNK2 in the mouse mammary epithelium promotes genetic instability and initiates extensive tumor development [66]. Likewise, deletion of MKK7, the other of the upstream MAPKKs for JNK, enhances the growth of breast and lung tumors in mice by impeding JNK mediated stabilization of the TP53 tumor suppressor protein [67]. On the other hand, the JNK pathway is often hyperactivated in various leukemias and solid tumors including cancers of the skin, lung, colon, brain, and liver [64,65], implicating JNKs as tumor promoters. For instance, the deletion of JNK2 or MKK4 in keratinocytes rendered mice resistant to skin cancer induction by a classic two-stage model that combines chemical mutagenesis of HRAS with proinflammatory phorbol ester treatment [68,69]. Likewise, JNK1 knockout strongly diminished the susceptibility of mice to chemically induced gastric cancer [70]. This dual effect of JNK on carcinogenesis is epitomized in hepatocellular carcinoma. On the one hand, JNK1 is hyperactivated in hepatocellular carcinoma and stimulates hepatocyte proliferation by downregulating expression of the cell cycle inhibitor CDKN1A [71]. On the other hand, knocking out both JNK1/2 in hepatocytes increases tumor growth, which is driven by enhanced cell death and compensatory hyperproliferation of surviving hepatocytes. Furthermore, proinflammatory cytokine production driven by JNK signaling in liver immune cells contributes to this phenotype by entertaining a permissive tumor microenvironment [72]. Thus, the exact function of JNK is tissue- and context-dependent, and to understand and exploit the role of this pathway in drug sensitivity and resistance, we need to understand this context. The literature on this topic is vast and sometimes contradictory, but some common themes seem to emerge. Thus, rather than trying to exhaustively review conflicting details, we focus on elaborating common themes through illustrative examples. In short, JNK can antagonize tumor growth by activating apoptotic pathways, or foster tumor aggressiveness and drug resistance by promoting cancer stem cell renewal, cell migration and cancer cell survival. Below we discuss these mechanisms considering tumor and tissue type diversification of these functions.

### 3.1. JNK Enhancing Resistance to ERK Pathway Inhibitors and Chemotherapeutic Drugs

The JNK pathway is in terms of the mechanism one of the best characterized escape routes for cells treated with RAF or MEK inhibitors. First, ERK and JNK have similar consensus phosphorylation motifs and share several substrates [73,74]. One of these shared substrates is the CJUN transcription factor, which promotes growth and survival of melanoma cells and allows cells to withdraw to quiescence and escape cell death when exposed to the RAF inhibitor vemurafenib. Consequently, JNK inhibition synergizes with vemurafenib to induce apoptosis [22]. Furthermore, the same work showed that vemurafenib-resistant cells retain high activity of the mTORC1 pathway, a key regulator of protein translation, which promotes survival and can be downregulated by JNK inhibition. Although it is not clear how the JNK and mTORC1 pathways cross-talk, these findings were confirmed and also shown to mediate resistance to joint BRAF and MEK inhibition in melanoma [21]. In the latter work, PAK was identified as kinase that reactivates the ERK pathway by phosphorylating CRAF in RAF inhibitor resistant cells, whereas PAK activates JNK in cells resistant to combined RAF and MEK inhibitor treatment.

Therefore, JNK clearly is important in mediating resistance to RAF inhibitors. However, what is the molecular mechanism? Again, nature’s cornucopia is plentiful and the mechanism seems to depend on how JNK is activated. Broadly speaking we can distinguish mechanisms based on external and internal signals. For instance, RAF inhibitor resistant melanoma cells can upregulate the expression of neuropilin-1 (NRP1), which uses the JNK pathway to induce the expression of receptor tyrosine kinases (RTKs), specifically epidermal growth factor receptor (EGFR/ERBB1) and insulin-like growth factor receptor 1 (IGFR1), that can bypass the signaling block imposed by RAF inhibition [75]. The same mechanism applies in breast cancer cells treated with HER2 targeting drugs. This mechanism could be relevant for selecting breast cancers sensitive to MEK inhibitor therapy. MEK inhibitors activate JNK by suppressing the expression of the JNK phosphatase DUSP4, which leads to the activation of ERBB RTKs in part through their enhanced transcription mediated by JUN. Consequently, breast cancers with loss of function mutations in the JNK pathway kinases MEKK1 or MKK4 should benefit from MEK inhibitor therapy, a hypothesis successfully confirmed in patient derived xenografts [76]. Similar mechanisms are found in glioblastoma, where EGFR inhibition induces secretion of tumor necrosis factor alpha (TNFα), which subsequently activates JNK, ERK, and AXL RTK [77]. Another stimulus to activate JNK can be supplied by the extracellular matrix (ECM). Triple-negative breast cancer cells grown on stiff, collagen-rich ECMs activate JNK through β1 integrin signaling and consequently become resistant to sorafenib, a RTK and RAF inhibitor [78]. Vice versa, JNK stimulates the expression of ECM components, such as osteopontin and tenascin C, which contribute to matrix stiffness and the establishment of metastatic niches [79]. JNK also may directly promote breast cancer cell survival in response to microtubule targeting agents, such as taxol, by phosphorylating BCL2 and BCLX proteins which protect mitochondrial integrity and counteract apoptosis [80]. Likewise, JNK sustains the expression of pro-survival proteins in lung cancer, thereby increasing resistance to cisplatin treatment [81].

However, the JNK pathway also responds to internal signals, e.g., to unfolded protein stress, which is commonly observed in cancer cells due to their high rates of protein synthesis [82,83]. Genetic screens identified the unfolded protein response sensor kinase IRE1 as a critical element to sustain the viability of mutant KRAS colorectal cancer (CRC) cells and as a key modulator of the response to MEK inhibitors [84]. Inhibitors of JNK, and its upstream activator MAPKKK TAK1, synergistically enhanced the efficacy of MEK inhibitors, suggesting that JNK also is intimately involved in coordinating internal cell stress responses with responses to anticancer drugs. In the same vein, JNK also can modulate resistance to 5-fluorouracil (5-FU) in CRC by enhancing BCL2 phosphorylation and autophagy, which protect against 5-FU induced apoptosis [85]. Such modulatory effects of JNK on responses to chemotherapies and targeted therapies were also observed in gastric cancer [86], hepatocellular carcinoma [87], head and neck cancer [88], endometrial cancer [89], and ovarian cancer [90]. An interesting finding in this context is that JNK signaling seems specifically important for the maintenance of cancer stem cells in different cancer types including pancreatic [91], endometrial [92], and ovarian cancer [93]. In ovarian cancer, the mechanism involves upregulation of survivin, an antiapoptotic protein that protects stem cell viability [94]. In summary, these results support a key role for the JNK pathway in mediating resistance to various targeted and non-targeted treatments that rests on JNK promoting survival signaling either directly by phosphorylating and inactivating apoptosis-regulating proteins, or indirectly by enhancing signaling through other—mainly RTK-mediated—survival pathways.

### 3.2. The Role of the JNK Pathway in Drug Sensitivity

The findings presented above make a convincing case that JNK promotes drug resistance. However, as always in biology, there is a flip side of the coin, i.e., situations where JNK enhances drug sensitivity. The targets are similar as in the resistance scenario, making it difficult to deduce general rules as to when JNK promotes drug resistance versus sensitivity. Plausibly, the differences may lie in different pathway activation kinetics as seen in the classic example of PC12 cells, where a transient activation of ERK induces cell proliferation while a more sustained ERK activation promotes cell differentiation [95]. In the absence of detailed kinetic comparisons this remains speculation, but arguably one worthwhile pursuing, as solving this conundrum will be necessary to decide when and where to deploy JNK inhibitors in cancer therapy.

Most of the evidence for a positive role of JNK in enhancing drug responses seems to rest on the modulation of proapoptotic proteins and phosphatases that regulate MAPK (including JNK) signaling. For instance, in ovarian cancer cells insulin followed by cisplatin treatment stimulates JNK activation, which leads to the accumulation of the TP53 tumor suppressor protein and increased cell death [96]. On the other hand, in lung cancer cells, JNK can reduce TP53 abundance by CJUN mediated repression of TP53 gene transcription resulting in reduced apoptosis and enhanced cisplatin resistance [97]. However, JNK also can have opposite effects in lung cancer. The micro-RNA miR-940 downregulates the expression of the JNK phosphatase DUSP1 and enhances cisplatin induced JNK signaling and cell death [98]. The expression of miR-940 is under negative control of RIP1, a TNFα receptor binding protein, and the authors speculate that interference with the RIP1/miR-940/DUSP1/JNK pathway could sensitize lung cancer cells to cisplatin. Similarly, in breast cancer DUSP1 is often overexpressed and is associated with poor prognosis and risk of relapse. Mechanistically, DUSP1 overexpression confers resistance to taxanes and anthracyclines by suppressing JNK activity [99]. Thus, the effects of JNK are highly context dependent, and to deploy JNK inhibitors for cancer treatment a thorough understanding of the network context in different cancer entities and even individual patients will be needed.

## 4. The Role of the p38 Pathway in Drug Resistance

The p38 pathway features a complex regulation. In addition to the usual activation by two upstream MAPKKs—MKK3 and MKK6—the p38 MAPKs can also be activated by binding of TAB1 (TAK1 binding protein 1) or phosphorylation by the tyrosine kinases ZAP70 and LCK. Both types of events trigger p38 autophosphorylation on activating residues. In addition, p38 comes in four flavors, p38α, p38β, p38γ, and p38δ, encoded by four different genes, whose protein products have overlapping and isoform specific functions [100,101,102,103]. The p38 pathway responds mainly to environmental stress signals and inflammatory stimuli and plays important roles in maintaining cellular homeostasis in different tissues including the neuronal and cardiovascular systems, but also in cancer. Simplified, p38 signaling protects cells against noxae, but promotes their demise when the damage sustained is persistent. This Janus-faced role is especially pronounced in cancer. p38α can counteract RAS transformation by suppressing ERK and JNK signaling, inducing cell cycle arrest and senescence [104,105,106,107,108]. Furthermore, p38α can restrain tumorigenesis in breast, lung, colon, liver, and potentially other tissues. A critical mechanism is the production of reactive oxygen species (ROS) in response to oncogene activation, which induces p38-mediated proapoptotic signaling and the killing of incipient tumor cells [109]. This suppression seems confined to tumor initiation. However, once a tumor is established p38 activity supports its growth [110]. A similar dichotomy applies to p38′s role in metastasis. In breast cancer, p38 can counteract bone metastasis by decreasing the expression of receptor activator of nuclear factor-κB (RANK) on tumor cells [111], which facilitates bone metastasis [112,113]. In CRC, p38 signaling downregulates the expression of parathyroid hormone-like hormone (PTHLH), which promotes lung metastases [114]. On the other hand, p38 promotes lung metastasis of breast cancer cells by upregulating pro-metastatic genes [115] and also epithelial-to-mesenchymal transition and metastasis in other cancers [116]. Thus, p38 directed tumor therapies may be a double-edged sword. Below, we will discuss the modulation of drug resistance and sensitivity by p38 signaling in more detail.

### 4.1. p38 Enhancing Resistance to Targeted Therapies and Chemotherapeutic Drugs

The anti-estrogen tamoxifen is a mainstay in the treatment of hormone sensitive breast cancers, and p38 pathway activation correlates with tamoxifen resistance and poor prognosis [117]. The activity of the p38 pathway is entertained by autocrine stimulation of breast cancer cells with vascular endothelial growth factor (VEGF) and pharmacologic p38 inhibition could alleviate tamoxifen resistance [118]. The role of RTKs in tamoxifen resistance is further supported by findings that ERBB1/2 expression and p38 activity were elevated in tamoxifen resistant breast cancer cell xenografts [119]. ERBB1/2 inhibition by gefitinib reduced p38 activity and tamoxifen resistance. Interestingly, p38 also modulates response to anti-hormonal therapies in prostate cancer. Initially, targeting the androgen dependency of prostate cancer cells is a useful strategy, but the eventual emergence of androgen-independent clones limits its long-term success. In these tumors, resistance to androgen deprivation therapy (ADT) is caused by p38 signaling to the FOXC2 transcription factor, and p38 inhibition restores sensitivity to ADT [120]. Similarly, the acquisition of an aggressive phenotype and resistance to zoledronic acid (a drug used to treat complications of bone metastases) are driven by p38 activation [121]. These observations point to an intimate relationship between hormone dependence and p38 signaling, which needs to be further elucidated if we want to exploit it for breaking resistance to hormonal based therapies. They are mainstay for the two most prevalent cancer types, i.e., prostate and breast cancer, and any advance in addressing resistance could have huge benefits for patient outcomes and modalities of healthcare provision.

Importantly, the role of p38 in drug resistance seems to intimately involve the tumor microenvironment. For instance, prostate cancer actively recruits mast cells, which increase resistance to taxane based chemotherapy and radiation therapy [122]. This study showed that mast cells increased drug resistance through expression of the CDKN1A cell cycle inhibitor mediated by p38 signaling to the TP53 tumor suppressor gene. A similar mechanism operates in breast cancer. Cancer associated fibroblasts (CAF) play an important part in sustaining and protecting tumor cells and may lend themselves to new unconventional approaches to cancer therapy [123]. CAFs co-cultured with breast cancer cells produce high levels of CCL11 and CXCL14 chemokines, which stimulate cancer cell proliferation and resistance to paclitaxel by activating p38 - STAT1 signaling [67,124]. Triple-negative breast cancer often shows overexpression of interleukin-1 receptor-associated kinase 1 (IRAK1), and paclitaxel treatment activates IRAK1, which increases the stemness of cancer cells and confers resistance to paclitaxel treatment. This resistance is mediated by p38 increasing the abundance of MCL1, an antiapoptotic member of the BCL2 family. Blocking IRAK1 or p38 activation restores paclitaxel sensitivity [125]. Although the source of p38 activation can differ and originate from inside or outside the cell, both intrinsic and acquired resistance to targeted treatment of HER2-overexpressing cancers involves p38 activation. Inhibiting p38 restored sensitivity to trastuzumab (Herceptin), a widely used biological to treat HER2 expressing breast cancers [126]. This study identified autocrine stimulation by growth differentiation factor 15 (GDF15) as the source of p38 activation. GDF15 also augmented the invasive potential of cancer cells, and the analysis of breast cancer tissues showed a significant relationship between p38 activation and cisplatin resistance and increased invasiveness in pancreatic cancer [127]. However, in this scenario, p38 activation was the cause of GDF15 overexpression suggesting that a positive feedback between GDF15 and p38 may exist that could drive drug resistance and invasiveness in different cancer types. GDF15 secretion by tumor cells generates an immune suppressive environment that accelerates the development of pancreatic cancers in animal models [128]. CAFs in the tumor microenvironment also contribute to 5-FU and oxaliplatin resistance in CRC. They produce growth factors that activate p38 and STAT3 in CRC cells, and p38 inhibition can reverse drug resistance [129].

However, CRC is also a good example for p38′s dual role. Inhibition of p38 can stimulate or inhibit CRC growth. This differential response is determined by the expression of the phosphatase PP2AC, which can correctly predict the response to p38 inhibitor treatment [130]. Mechanistically, PP2AC controls the balance of signaling between the p38 and ERK pathways. Low PP2AC expression levels diminish the inhibition of ERK by p38, allowing ERK to induce TSC2 phosphorylation at inhibitory sites (S664, S1798), which allows mTORC1 to become activated. Conversely, when PP2AC expression is high, p38 via its downstream effector kinase MK2 phosphorylates TSC2 on an activating site (S1254) resulting in mTORC1 inhibition. Thus, p38 inhibitors increase mTORC1 signaling and stimulate proliferation in cells with high PP2AC expression, whereas they have the opposite effects in cells with low PP2AC expression. Combination of a p38 with an mTORC1 inhibitor could overcome the resistance to p38 inhibition causing a marked suppression of tumor growth in animal models. These results elegantly show that a detailed understanding of the wiring of signal transduction networks is key to the successful deployment of targeted therapies and the design for efficacious combination treatments.

In non-small cell lung cancer (NSCLC), the EGFR inhibitor gefitinib can achieve remarkable responses, but also induce tetraploidization which leads to p38 mediated drug resistance. Pharmacological inhibition of p38 prevented tetraploidization and could restore gefitinib sensitivity in PDX models [131]. However, p38 also plays an important role in chemotherapy resistance in lung cancer. Cisplatin resistance and p38 signaling in NSCLC can be increased by the transcriptional regulator Inhibitor of differentiation 4 (ID4) [132]. Transcriptional regulation also plays a role in NSCLC resistance to paclitaxel. Resistant cells upregulate EGFR and p38 signaling leading to the stabilization of TP53, the transcription factor responsible for enhanced EGFR production [133].

In part, p38 may induce chemotherapy resistance by enhancing stemness in cancer cell populations. Cisplatin-resistant lung cancer cell spheres adopted properties of CSCs and resilience to apoptosis, which was conferred by p38 phosphorylating the cytoprotective heat shock protein HSP27 [134]. Knockdown of either p38 or HSP27 enhanced cisplatin sensitivity. In head and neck cancer patients, the upregulation of p38 expression and activation in the surgical resection margin of the tumor is associated with the expression of CSC markers (OCT4, KLF4, MYC, and CD44) and a higher frequency of relapse [135]. Treatment of cultured tumor cells with p38 inhibitors reduced the expression of CSC markers and sensitized the cell to DNA damage induction and cisplatin.

### 4.2. The Role of the p38 Pathway in Drug Sensitivity

As p38 can efficiently trigger apoptosis pathways [101,110], it is no surprise that it can also execute and enhance the effects of cancer drugs. As the examples with CRC show [101,130], these effects are highly context dependent, but a common theme seems to emerge in leukemia and other malignancies. The p38 pathway can cause resistance to chemotherapeutic leukemia treatments through a variety of mechanisms including induction of autophagy, upregulation of drug efflux transporters, and cell cycle regulation that helps cells to survive the damage caused by these largely genotoxic drugs [103]. However, when leukemia is treated with targeted therapies, such as chronic myelogenous leukemia (CML) with tyrosine kinase inhibitors, p38 seems to change allegiances and promotes the effects of these targeted therapies. CML is driven by the BCR-ABL oncogene: a deregulated tyrosine kinase created by a fusion between the BCR and ABL genes. Imatinib, the first clinically used BCR-ABL inhibitor, revolutionized the treatment of this fatal disease and turned it into a manageable entity [136]. Imatinib not only blocks BCR-ABL kinase activity, but also activates p38 signaling which is necessary for imatinib’s growth suppressing effects [137]. Similarly, dasatinib, a tyrosine kinase inhibitor effective against imatinib resistant BCR-ABL mutants, uses the p38 pathway to induce growth arrest, apoptosis and anti-leukemic responses [138].

The p38 pathway also may counteract drug escape mechanisms. PI3K (phosphoinositide-3-kinase)/mTOR inhibitors are initially successful in suppressing tumor growth in animal models of ovarian cancer, but eventually resistance occurs due to the upregulation of the MYC and YAP transcriptional regulators [139]. This upregulation requires ERK activity and inhibition of p38 and was also observed in other cancer cells carrying KRAS, BRAF, or NF1 mutations, suggesting that p38 antagonizes this resistance pathway in several different cancer types. Furthermore, p38 signaling may be involved in antagonizing resistance pathways in cells that are already drug resistant. For instance, p38 inhibition further increased carboplatin resistance in cisplatin-resistant ovarian cancer cells, but not in sensitive cells [140]. The same study showed that p38 inhibition enhanced the viability of primary ovarian cancer cells obtained from ascites, indicating that basic p38 signaling restrains cell survival and its inhibition may be necessary to expand the malignant cell population. Thus, p38 may impact cancer cell survival and drug resistance by modulating other pathways. In breast cancer, resistance to tamoxifen can be caused by overexpression of phosphodiesterase 4D (PDE4D) [141]. Investigating the mechanism revealed that inhibiting PDE4D restored tamoxifen sensitivity through induction of the unfolded protein response and subsequent activation of p38 and JNK. In lung cancer, blocking the downstream effectors MEK or PI3K is ineffective in overcoming resistance to EGFR inhibitors [142]. However, combined MEK and PI3K blockade successfully interfered with features of malignant transformation caused by drug-resistant EGFR mutants. Mechanistically, combination treatment activated p38 signaling and apoptosis. This could be an interesting paradigm for combinations of targeted therapies recruiting p38 to enhance their efficacy by adding apoptosis promotion.

## 5. Metabolism, MAPK Signaling, and Drug Resistance

Rewiring of cell metabolism and bioenergetics is a characteristic feature of cancer development and has been recognized in recent years as a hallmark of cancer [143]. Metabolic alterations in cancer were first identified in the 1920′s by Otto Warburg, who described the upregulation of glycolysis and increased lactate production in tumor cells under conditions of sufficient oxygen availability. This particular metabolic phenotype is now known as aerobic glycolysis or the “Warburg Effect” [144]. This adaptation is beneficial to cancer cells as it supports the demand for rapid ATP synthesis and fuels biosynthetic pathways to meet the requirements of rapidly proliferating cells [145]. However, the extent of the benefits these metabolic alterations provide to cancer cells is still not fully understood, particularly for drug sensitivity and resistance.

### 5.1. The Role of Glycolysis in Drug Sensitivity and Resistance

Hardeman et al. described the variability in the response of BRAF-mutated melanoma cells in vitro to the RAF inhibitor PLX4720 [146]. They identified a relationship between drug sensitivity and glycolytic dependence in different cell lines and showed that diminishing mitochondrial activity through mitochondrial DNA depletion could attenuate intrinsic RAF inhibitor resistance. This study highlights the therapeutic potential of targeting metabolic vulnerabilities, particularly in patients who show no response to targeted therapy. Further studies have identified a clear relationship between the presence of the mutant BRAF V600E protein and elevated glycolytic activity in various cancers [147,148,149]. Parmenter et al. demonstrated that the suppression of glycolysis through BRAF inhibition in melanoma cells correlated with the downregulation of glucose transporters, GLUT1/3, and hexokinase 2 (HK2) expression [150]. This was also observed in biopsies from melanoma patients harboring BRAF V600E mutations. However, ectopic expression of the NRAS Q61K mutant induced resistance to RAF inhibitors and restored glycolytic activity in these cells. Combining the RAF inhibitor vemurafenib with the glycolytic suppressor, dichloroacetate resensitized the vemurafenib-resistant melanoma cells to BRAF inhibition. Furthermore, microarray analysis identified BRAF V600E as a regulator of several transcriptional regulators of glycolysis, including MONDOA, HIF-1a, and MYC [150]. Although the resistant cells used in this study had gained an activating NRAS mutation, other acquired resistance mechanisms may also be susceptible to treatments co-targeting glycolysis, if they present similar metabolic profiles or dependencies.

### 5.2. The Influence of Glycolytic Enzymes on MAPK Signaling in Cancer

The links between MAPK signaling and metabolic processes in cancer have been widely studied; however, the extent of the roles played by individual MAPK pathway components in metabolic regulation is poorly understood. Similarly, the potential functions of metabolic proteins in MAPK pathway regulation are just as elusive. Fructose-1,6-bisphosphatase (FBP1) is the rate-limiting enzyme controlling gluconeogenesis and glucose homeostasis. Studies showed a correlation between low FBP1 expression and higher aggressiveness and worse prognosis in cancer patients [151]. Jin et al. discovered that FBP1 could inhibit the interaction of the scaffold protein IQ-domain GTPase-activating protein 1 (IQGAP1) with ERK1/2, thereby hindering IQGAP1-mediated activation of ERK1/2 in pancreatic ductal adenocarcinoma (PDAC) cells [152]. Gemcitabine—a standard chemotherapeutic drug used in PDAC treatment—induces ERK phosphorylation in pancreatic cancer cells, which can lead to resistance development. By combining gemcitabine treatment with a FBP1-derived peptide inhibitor, gemcitabine-induced ERK activation could be blocked [152]. This indicates that FBP1 can exert signaling tasks separate to its enzymatic metabolic functions, which could be an exploitable therapeutic target for pancreatic cancer patients.

Pyruvate kinase M2 variant (PKM2) is a glycolytic enzyme converting phosphoenolpyruvate (PEP) to pyruvate, but was also shown to bind to various signaling kinases, such as ARAF [153] and ERK1/2 [154]. Oncogenic ARAF can directly bind to PKM2 and induce tetramerization, thereby increasing the activity of PKM2 [153]. When in a complex with SAICAR (succinyl-5-aminoimidazole-4-carboxamide-1-ribose-5′-phosphate), PKM2 also can bind to and activate ERK1. In turn, activated ERK can phosphorylate PKM2 and sensitize it for SAICAR-binding. These positive feedback loops support proliferative signaling in cancer cells [154]. The ability of metabolic proteins to bind to other signaling pathway components and regulate their activity suggests a new range of potential targetable roles in MAPK pathway regulation and oncogenic signaling in cancer.

### 5.3. The Role of Amino Acid Metabolism in Cancer and Drug Resistance

As cancer cells undergo rapid proliferation, they require a constant supply of building blocks to sustain tumor growth. Non-essential amino acids (NEAAs) play a significant role in protein synthesis and can also be utilized as carbon and nitrogen sources for nucleic acid and lipid synthesis. In addition, NEAAs can provide energy and defense against oxidants, and therefore cancer cells can become dependent on specific amino acids [155]. Ross et al. demonstrated a potential role of serine biosynthesis in acquired resistance to BRAF inhibitors [156]. Proteomic analysis revealed that the abundances of key enzymes in the serine biosynthetic pathway, i.e., D-3-phosphoglycerate dehydrogenase (PHGDH), phosphoserine phosphatase, and phosphoserine aminotransferase 1, were significantly reduced by vemurafenib treatment of sensitive wild type SK-MEL-28 cells, but not in vemurafenib resistant cells. Furthermore, they showed that siRNA knockdown of PHGDH along with serine depletion could effectively resensitize the resistant cells to vemurafenib treatment [156]. The role of amino acid metabolism in drug sensitivity and acquired drug resistance is marginally understood, but it is apparent that cancer cells depend on a variety of amino acids, such as arginine [157], proline [158], and aspartate [159] for tumor survival under various conditions.

### 5.4. The Role of Mitochondrial Metabolic Rewiring in Drug Sensitivity and Resistance

Warburg proposed mitochondrial dysfunction as a cause of cancer metabolic reprogramming. However, this was deemed incorrect as most tumor mitochondria can effectively carry out oxidative phosphorylation and can also rewire their metabolism to promote biosynthesis and survival under stress [160]. Several studies have investigated the metabolic response of tumor cells in stressful environments, such as drug-induced pressure. Cesi et al. found that an increase in ROS production led to inhibition of pyruvate dehydrogenase (PDH) in response to BRAF inhibitor treatment in BRAF V600E mutated, but not wild type BRAF or NRAS mutated melanoma cells. This was associated with activated pyruvate dehydrogenase kinases (PDKs) as a survival mechanism to reduce pyruvate flow to the citric acid cycle and limit oxidative metabolism, thereby lowering the detrimental ROS production. They showed that the PDK inhibitor AZD7545 increased ROS levels and could suppress the growth of both BRAF V600E-mutated and BRAF inhibitor-resistant cells [161]. This suggests that PDKs could be a potential metabolic target in cancer therapy. However, this study also highlights the variability in metabolic rewiring between melanomas with different oncogenic drivers and suggests that mutant BRAF and mutant NRAS may utilize different metabolic pathways for survival benefits.

Although it is well established that increased glycolysis and a dependency on the Warburg Effect are commonly observed in BRAF driven melanoma cells [162], recent studies on resistance to BRAF inhibitors in melanoma suggest a greater role of mitochondrial metabolism in a subset of cells. Vazquez et al. described the effects of peroxisome proliferator-activated receptor gamma coactivator 1-alpha (PGC1a) levels on metabolic programming and drug sensitivity in melanoma. They found that a subset of melanoma cells with melanocyte inducing transcription factor (MITF)-driven overexpression of PGC1a display increased mitochondrial oxidative metabolism and ROS detoxifying abilities to withstand significant oxidative stress. In contrast, melanoma cells with low expression of PGC1a favor glycolysis and are more susceptible to high ROS levels, indicating that these cells could potentially be targeted by ROS-inducing drugs [163]. Baenke et al. demonstrated that BRAF inhibitor-resistant melanoma cells can also have an increased dependency on mitochondrial metabolism, which contributes to their resistance. The enhanced oxidative phenotype was a result of utilizing predominantly glutamine instead of glucose as carbon source. This was associated with an increase in glutamine uptake and an upregulation in the expression of glutaminase, an enzyme that generates glutamine from glutamate [164]. Targeting glutaminolysis and other mitochondrial functions is therefore of potential therapeutic interest for cancers displaying oxidative metabolic vulnerabilities.

### 5.5. The Potential Role of Fatty Acid Metabolism in MAPK Signaling and Drug Resistance

Aerobic glycolysis and increased rates of energy-demanding biosynthesis of macromolecules are the two most common metabolic characteristics of cancer cells. However, another feature common to virtually all cancers is an upregulation of de novo fatty acid (FA) synthesis. This is associated with glycolysis as the glycolytic pathway supplies the energy and fuel required for FA synthesis [165]. Although glycolysis and mitochondrial metabolism have been extensively studied in cancer and drug resistance, the role of lipid and fatty acid metabolism has become an emerging topic of interest. Feng et al. demonstrated the reprogramming from de novo FA synthesis to exogenous FA uptake in acquired HER2 inhibitor resistance in breast cancer [166]. cDNA microarrays indicated the upregulation of the CD36 receptor, a FA transporter, in lapatinib (dual ERBB1/2 inhibitor)-resistant cells. They showed that inhibition of CD36 could selectively suppress the growth of lapatinib resistant but not lapatinib sensitive cells, both in vitro and in mouse models. Deletion of the CD36 gene in a mouse model of breast cancer significantly attenuated mammary tumor development [166]. These results suggest that CD36 along with other FA transporters may play a role in the acquired resistance mechanisms of cancer cells. A study previously identified the activation of SRC, and subsequent ERK1/2 activation, through CD36-mediated oleic acid uptake in cervical cancer [167]. Thus, CD36 may promote tumor progression by cross-talking with signaling pathways such as ERK, and therefore targeting CD36 in cancer could be a promising strategy.

In summary, it is evident that while tumors may have the same oncogenic drivers, their metabolic phenotypes may be influenced by a combination of factors, such as altered metabolic gene expression and pathway up- or downregulation. By elucidating the metabolic dependencies of tumors, it may become possible to target unique metabolic profiles of different cancer types in addition to their driver oncogenes. Using metabolic combination therapies could potentially limit the acquisition of secondary mutations and prevent or delay treatment insensitivity.

## 6. Epigenetics and Drug Resistance

Although cancer is usually considered a genetic disease, it has long been recognized that malignancies are also characterized by epigenetic aberrations. These dynamic mechanisms include genome-wide DNA methylation, chromosome structures, nucleosomal positioning, and post-translational histone modifications. Together, they allow the cell to encode highly distinct and precise gene expression profiles and adaptive cell phenotypes during cellular development and diseases such as cancer. In healthy cells, the distinct organization of the chromatin regulates and tightly controls the way signaling pathways and transcription factors alter gene expression activity resulting in distinct cellular phenotypes. This so-called epigenetic homeostasis is crucial for the cellular identity and cell state. This chromatin homeostasis is disrupted in cancer either through genetic stimuli, for instance mutations in chromatin regulatory genes, or non-genetic stimuli, e.g., hypoxia and inflammation. Similar to genetic aberrations, epigenetic changes directly contribute to genetic instability and a mutant phenotype, and are therefore considered as emerging hallmarks in cancer [143]. Importantly, however, epigenetic plasticity and adaptations are major contributors in the way tumor cells respond to therapies. The role of epigenetics in cancer initiation and progression was extensively reviewed [168,169,170,171,172,173,174,175,176,177]. Therefore, we will mainly focus on how epigenetic changes and factors influence resistance and sensitivity to drugs in MAPK mediated malignancies.

It has long been established that, in addition to classical genetic mutations, epigenetic factors also contribute to tumor heterogeneity and drug resistance. During the process of oncogenesis, genetic as well as epigenetic instabilities lead to the development of tumor cells. As tumors develop over time, the parental tumor cell might acquire new mutations and undergo epigenetic alterations leading to the development of subclones with different functional repertoires for tumor growth, metastasis, or drug tolerance. Overall, both genetic and epigenetic factors contribute to the development of phenotypically distinct tumor cell clones and intratumoral heterogeneity [168,178,179].

In line with the previous chapters, the combination of genetic and epigenetic alterations has implications for the treatment of cancers and, importantly, contributes to drug resistance. In addition to signaling network adaptations and resistance mutations in a targeted molecule, epigenetic alterations seem to play similarly important roles in drug resistance. Due to the extensive characterization of acquired RAF inhibitor resistance in malignant melanoma, the involvement of epigenetic mechanisms and resistance to targeted therapies is best understood in these cancers [180].

### 6.1. Epigenetics and Resistance to MAPK Pathway Inhibition

Approximately 10 percent of human malignancies are driven by gain-of-function mutations in the BRAF oncogene resulting in constitutive activation of the RAS-RAF-MEK-ERK pathway [12]. Despite initial high response rates with targeted therapies against RAF and MEK, these therapies are limited due to the emergence of drug resistance. In malignant melanoma, BRAF- and MEK inhibitor-tolerant cells exhibit multiple transcriptional states [181] and, over the last years, a number of acquired resistance mechanisms were discovered (reviewed in [12,13] and Figure 3). In addition to genetic and non-genetic alterations a number of epigenetic mechanisms were identified that can drive acquired resistance and insensitivity to drug treatment.

In a cellular model for acquired vemurafenib resistance in BRAF V600E mutated melanoma, Gupta and colleagues systematically identified epigenetic regulators mediating drug resistance using a short-hairpin library to downregulate more than 300 genes known to be involved in epigenetic control [182]. Their data indicate that reducing the expression of the ribosome biogenesis protein Block of Proliferation 1 (BOP1) confers resistance to BRAF kinase inhibitors by downregulating the expression of the MAPK phosphatases DUSP4 and DUSP6, which de-activate ERK and both ERK and p38, respectively. ERK stimulates the transcription of both phosphatases, and the reduction of this feedback inhibition leads to activation of MAPK signaling and increased RAF inhibitor resistance.

The mutant BRAF V600E protein requires the chaperone HSP90 for folding and stability and consequently is degraded in response to Hsp90 inhibitors [183]. In colorectal and mammary cancer, the Ubiquitin-Specific Protease 22 (USP22) enhances the expression of HSP90AB1 and promotes resistance to HSP90 inhibition. USP22 is part of the deubiquitinating module (DUBm) of the SAGA (Spt-Ada-Gcn5 Acetyltransferase) complex that deubiquitinates the core histone H2B at lysine 120. USP22 depletion correlated with diminished binding of the SAGA complex to the HSP90AB1 gene promoter and increased sensitivity to HSP90 inhibition [184].

In a number of cancers, phenotype plasticity and drug tolerance is associated with altered expression levels of crucial histone-modifying enzymes. Hou and colleagues could demonstrate that oncogenic BRAF V6000E in melanoma causes a large number of epigenetic alterations, including decreased expression levels of DNA-methyltransferase 1 (DNMT1) and histone methyltransferase EZH2 affecting CpG island methylation and the epigenetic landscape [185]. Using a BRAFV600E zebrafish model, Ceol and colleagues demonstrated that a number of genes involved in epigenetic regulation were found to cooperate with the BRAF V600E driver oncogene to accelerate melanomagenesis [186]. Among others, the lysine methyltransferase SETDB1 known to methylate histone H3 lysine 9 (H3K9), was shown to accelerate melanoma formation. Interestingly, expression of SETDB1, as well as SETDB2, was also upregulated in other acquired resistance models for melanoma, lung, breast, and colon cancer [187]. Upregulation of the histone modifiers was observed in response to BRAF and MEK inhibition, whereas their knockdown restored drug sensitivity.

One cut homeobox 2 (ONECUT2), a transcription factor, is highly overexpressed in a number of human malignancies including prostate and lung cancer [188,189,190,191]. Overexpression of ONECUT2 in RAS-driven lung adenocarcinoma promotes tumor growth and invasion [191]. ONECUT2 activates distinct gene expression profiles by promoting the trans-differentiation of lung cancer cells by modulating Polycomb Repressive Complex 2 (PRC2) occupancy of distinct chromatin domains. Here, ONECUT2 seems to regulate H3K27me3 modification by controlling PRC2 occupancy.

Roesch and colleagues identified another intrinsic resistance mechanism in BRAF V600E mediated malignant melanoma mediated by altered histone modifications [192]. Exposure of BRAF V600E driven melanoma cells to sublethal levels to the BRAF inhibitor vemurafenib or the DNA damaging agent cisplatin led to the development of a less differentiated, drug-tolerant state characterized by the enrichment of slow-cycling, long-term tumor-maintaining melanoma cells. This phenotype switch was accompanied by increased expression levels of the H3K4-demethylase JARID1B/KDM5B/PLU-1 and an upregulation of mitochondrial oxidative phosphorylation. Importantly, inhibition of oxidative phosphorylation blocked the emergence of the cell population with elevated JARID1B and sensitized melanoma cells to therapy.

A recent study not only identified a novel epigenetic mechanism of drug resistance in BRAF V600E mediated melanoma, but also utilized this information for developing combinatorial therapies that could overcome drug induced resistance. Using a large-scale CRISPR-Cas9 gene knockout screen in cell models for resistance to RAF inhibition (Dabrafenib) alone or in combination with MEK inhibition (Trametinib), Strub et al. identified a role for the histone deacetylase SIRT6 [193]. Interestingly, only haploinsufficiency, but not the complete loss of SIRT6, led to resistance to BRAF or MEK inhibition. SIRT6 haploinsufficiency resulted in increased IGFBP2 expression due to increased chromatin accessibility and H3K56 acetylation at the IGFBP2 locus, which in turn resulted in IGF receptor activation and downstream AKT signaling. Targeting this novel node by combining IGFR inhibitors with Dabrafenib could overcome acquired resistance in SIRT6 haploinsufficient melanoma cells.

In another study, BRAF V600E expressing malignant melanoma cells with resistance to RAF inhibitors showed activation of the EGFR/PI3K/AKT pathway [194]. Interestingly, this bypass mechanism involved an increase in EGFR expression that was regulated by hypomethylation of CpG sites in the EGFR promotor region. Importantly, the authors could demonstrate that drug resistance in these cells could be overcome by combination treatment with EGFR and BRAF inhibitors.

### 6.2. BET Family of Epigenetic Regulators

The Bromodomain and Extraterminal Domain (BET) gene family, including BRD2, BRD3, BRD4, are so-called “histone readers” of lysine acetylation of the N-terminal part of histone and involved in regulating sensitivity to MAPK inhibition in a number of cancers. In several human cancer models BRD4 was shown to interact with the transcription factors YAP and TAZ, thereby controlling the genome-wide association of BRD4 to chromatin and expression of tumor growth-regulating genes [195]. Small-molecule inhibitors of BRD4 were able to suppress YAP/TAZ pro-tumorigenic activity and revert drug resistance.

Similarly, combining BET with BRAF inhibitors synergized to increase cell death and reduce tumor cell proliferation in BRAF V600E mutated melanoma [196,197]. Importantly, the same concept proved successful in mutant NRAS-driven malignant melanoma, where BRD4 and MEK inhibitors synergized to inhibit tumor growth in mouse melanoma models [196]. Combinatorial treatment with MAPK and BET inhibitors also could overcome intrinsic MAPK inhibitor resistance in colorectal cancer models [198]. Overall, the BET family of epigenetic regulators has emerged as promising therapeutic targets in MAPK mutated cancers and a number of small molecule inhibitors of BET proteins have been developed or undergoing clinical investigation.

### 6.3. Targeting Histone Deacetylases (HDACs)

Histone deacetylases (HDACs) are a large family of enzymes mediating the removal of acetyl groups from histone, thereby regulating histone-DNA interactions. HDAC expression is altered in a number of cancers and several HDAC inhibitors were tested for cancer treatment alone or in combination with other therapies [175,199].

A number of studies demonstrated that resistance to BRAF and MEK inhibitors in malignant melanoma is associated with increased ROS levels [200]. The study suggests that the HDAC inhibitor vorinostat suppresses the expression of the sodium-independent cysteine–glutamate antiporter SLC7A11 resulting in a lethal increase in cellular ROS levels in drug-resistant cells, thereby overcoming insensitivity to ERK pathway inhibition. Similarly, Maertens and colleagues demonstrated that both RAF and MEK inhibitor sensitive and resistant melanoma can be effectively treated with a combination of RAF/MEK inhibitors and the HDAC3 inhibitor entinostat [201]. Mechanistically, ERK pathway inhibition suppressed homologous recombination-based DNA repair, whereas entinostat suppressed nonhomologous DNA repair—a double assault that leaves the cell defenseless against DNA damage. Importantly, this combination also is effective in mutant NRAS driven melanomas, which are insensitive to MEK or RAF inhibitors. Similarly, HDAC8 expression was increased in malignant melanoma cell models of acquired resistance to BRAF inhibitors due to the deacetylation of the transcription factor CJUN, which promotes the subsequent upregulation of EGFR and ERK signaling [202]. Importantly, this study demonstrated that HDAC8-specific inhibitors are able to enhance the durability of BRAF inhibitor therapy. In a model of BRAF, mutated CRC increased c-MET-STAT3 signaling was identified as a novel adaptive resistance mechanism to MEK inhibition resulting in increased transcription of the endogenous caspase-8 inhibitor c-FLIPL [203]. HDAC inhibitors could efficiently suppress c-FLIPL expression augmenting MEK inhibitor-induced cell death. HDAC inhibition also proved successful in mutant KRAS pancreatic cancer [204]. Here, the treatment with the HDAC inhibitor MPT0E028 in combination with a MEK inhibitor yielded synergistic effects to defeat the intrinsic resistance to ERK pathway inhibitors.

Together, these studies highlight the phenomenon of non-oncogene addiction in tumors resistant to ERK pathway inhibitors [205,206]. Here, resistance mechanisms are often dependent on the expression of per se non-oncogenic genes that collaborate with the driver oncogenes to abrogate the inhibition of driver oncogene signaling. Frequently, these collaborator genes are not activated through genetic alterations, but by epigenetic mechanisms. In recent years, a number of studies suggested that drugs targeting epigenetic alterations could be applied in synergy with other anticancer therapies or, importantly, in reversing acquired therapy resistance (reviewed in [169,176,177,207,208]. However, due to the very limited tolerability of drugs targeting epigenetic mechanisms in combination with targeted therapies, most clinical trials have been disappointing so far. Dose adjustments, sequential scheduling and targeted delivery of drugs might overcome these hurdles and result in improved combinatorial treatment regimes.

## 7. Conclusions

Current evidence clearly shows that MAPK pathways are viable targets for cancer therapy. The most prominent and clinically utilized target is the ERK pathway. However, stress-activated MAPK pathways, such as JNK and p38, play important modulatory roles that can change the response of cancer cells to both targeted therapies and chemotherapies, and so do metabolic and epigenetic modulators. However, these roles are context dependent and difficult to foresee using traditional empirical discovery methods. Going forward we will need to increasingly rely on computational modeling methods that can analyze and interpret these complex relationships and predict solutions which are often non-obvious and non-intuitive [49]. The goal is to construct digital twins or computational avatars of cancer patients so that we can determine and test the best possible therapy for each individual patient in silico before we administer it to the real patient (Figure 5).

Is this a daydream? We argue that this is the future of oncology. This type of computational modeling is already widely and very successfully applied to extremely complex problems in engineering. Therefore, why not use it to step change cancer therapies. We will not even be able to test all reasonable drug combinations that consider and exploit the intricate relationships between MAPK pathways, let alone other pathways. We have neither enough time, money, nor patients to do this in traditional clinical trials. We need to do this in silico using computational models.

## Figures and Tables

**Figure 1 ijms-21-01102-f001:**
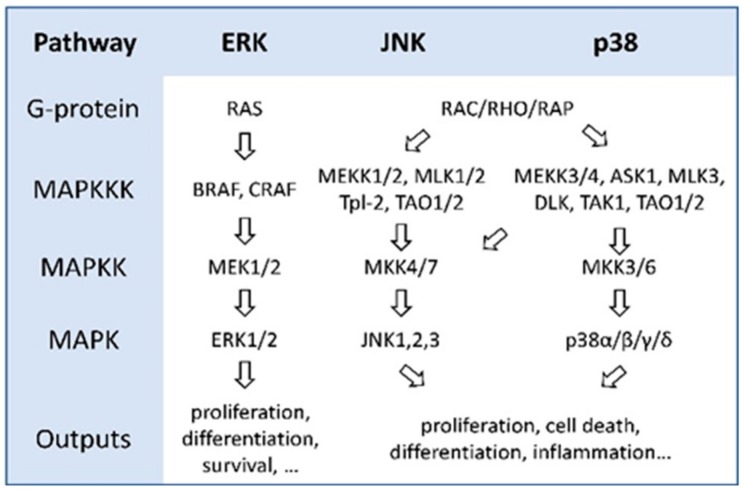
Organization of MAPK pathways. The MAPK core consists of three kinases (MAPKKK, MAPKK, and MAPK), which form a signal transduction cascade that receives input from G-proteins and produces different biological outputs.

**Figure 2 ijms-21-01102-f002:**
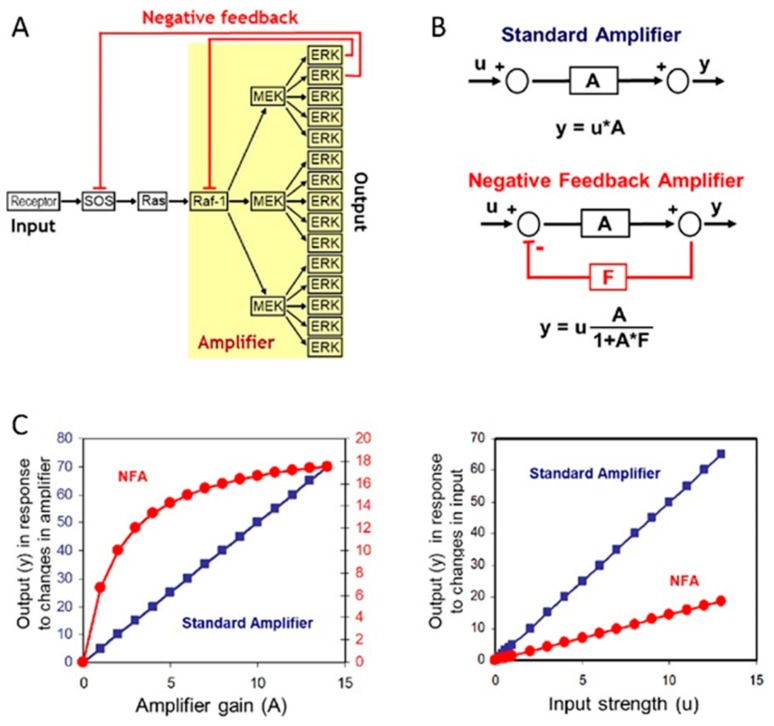
The ERK pathway functions as a negative feedback amplifier (NFA). (**A**) Schematic representation of the ERK pathway with approximate stoichiometries of pathway components typically found in cells and negative feedbacks indicated. (**B**) Comparison of a standard amplifier and NFA. The formula relating input (u) to output (y) shows that the NFA output is dominated by the strength of feedback (F) rather than the amplification (A). (**C**) Comparison of the standard amplifier (blue) and NFA (red). Figure adapted from [2].

**Figure 3 ijms-21-01102-f003:**
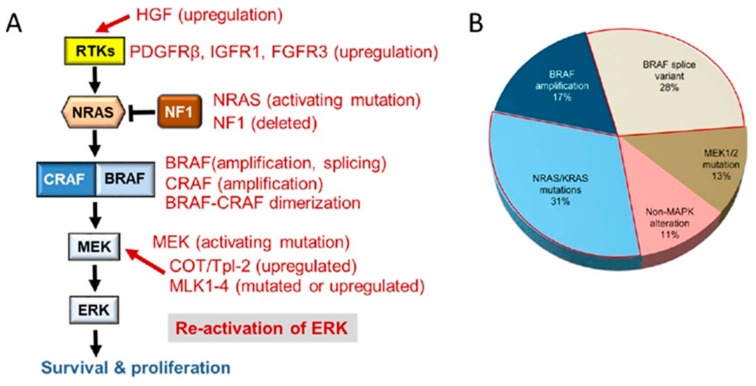
Resistance mechanisms to RAF inhibitor. (**A**) Different types of signaling network adaptations restore ERK activation. No mutations that compromise drug binding to the target (RAF) have been observed. (**B**) Mechanisms that enhance RAF protein dimerization account for ~60% of drug resistance [17].

**Figure 4 ijms-21-01102-f004:**
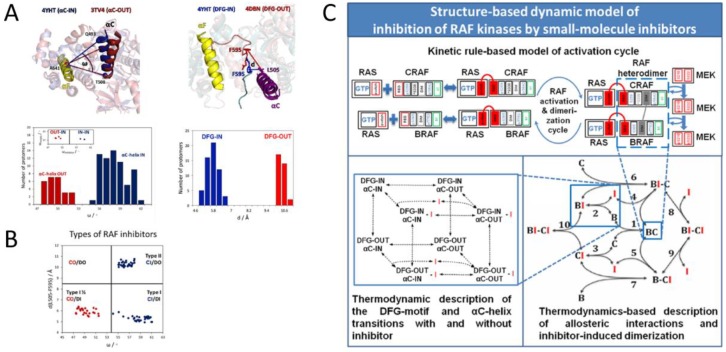
Targeting RAF dimerization. (**A**) X-ray structures showing the two conformations (in/out) of the Cα helix and DFG motif and their distribution in available RAF structures deposited in the Protein Data Bank. (**B**) Frequency of in/out distributions and RAF inhibitors targeting such conformations. (**C**) Depiction of the computational modeling approach developed by Rukhlenko et al. [57].

**Figure 5 ijms-21-01102-f005:**
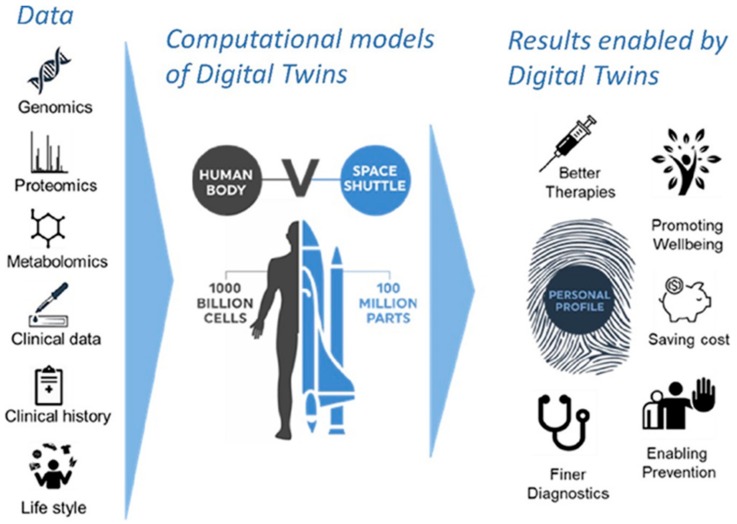
The Digital Twins concept. Integrating molecular, clinical, and imaging data through computational models will allow us to generate a digital twin of each patient. Treatments can be first tried on the digital twin and the best option selected for the treatment of the real patient. Likewise, personal prevention pathways can be designed. Both approaches optimize patient outcomes while minimizing costs as patients will not receive unnecessary treatments.

**Table 1 ijms-21-01102-t001:** The mechanisms of action and acquired resistance to MAPK inhibitor drugs. Note that MEK inhibitors are used in combination with BRAF inhibitors.

MAPK Inhibitor	Mechanism of Action	Potential Acquired Resistance Mechanism(s)
Vemurafenib	Selective BRAF V600E inhibitor (ATP competitive)	Activating mutations in NRAS or MEK [23,24]; BRAF/CRAF amplification [23,25]; BRAF alternative splicing [26,27]; RAF heterodimerization [28]; RTK upregulation [29]; Loss of NF1 [30]; COT upregulation [31]
Dabrafenib		
Encorafenib		
Trametinib	Selective MEK1/2 inhibitor (ATP non-competitive)	
Binimetinib		
Cobimetinib	Selective MEK1 inhibitor (ATP non-competitive)	
Gefitinib	Selective EGFR inhibitor (ATP competitive)	EGFR T790M mutation [32,33]; Gene amplification (e.g., MET and HER2) [34,35]; Loss of NF1 [36]; BRAF mutation [37]
Lapatinib	EGFR and HER2 inhibitor (ATP competitive)	Compensatory pathway activation (e.g., Akt, mTOR, HER3, and ER) [38,39,40,41]; Gene amplification (e.g., TRAPPC9) [42]; TK domain mutations in HER2 [43,44]

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
