# Peer review of "Targeting MAPK Signaling in Cancer: Mechanisms of Drug Resistance and Sensitivity"

_ijms, 2020, doi:10.3390/ijms21031102_

Round 1

Reviewer 1 Report

In the proposed manuscript, entitled “Targeting MAPK signaling in cancer: mechanisms of drug resistance and sensitivity”, Kolch et al. describe the main mechanisms of resistance to MAPK inhibitors in cancer, providing an overview in this field.

The overall manuscript structure appears well organized. In order to improve the manuscript quality, some minor changes should be accomplished.

Minor concern:

In order to have a comprehensive overview of these mechanisms, authors should include a table listing all the MAPK inhibitors, described in the manuscript, the respective mechanism of action and resistance and the reference. Images quality and size should be improved. In the text some typos are present and English should be revised (e.g. in Figure 3A the authors use the term “dimerisation” instead of “dimerization” present in the whole text). The authors should render the aim of the review more appealing to the readers. For instance, by modifying the sentences “For the remainder of the review we focus on discussing less well reviewed areas of MAPK signaling” lane 47-50 or “Thus, rather than trying to exhaustively review conflicting details we focus on elaborating common themes through illustrative example” lane 152-153.

Author Response

Response to Reviewer 1 (our responses are marked by >)

Comments and Suggestions for Authors

In the proposed manuscript, entitled “Targeting MAPK signaling in cancer: mechanisms of drug resistance and sensitivity”, Kolch et al. describe the main mechanisms of resistance to MAPK inhibitors in cancer, providing an overview in this field.

The overall manuscript structure appears well organized. In order to improve the manuscript quality, some minor changes should be accomplished.

> We would like to thank the reviewer for the overall very positive evaluation.

Minor concern:

In order to have a comprehensive overview of these mechanisms, authors should include a table listing all the MAPK inhibitors, described in the manuscript, the respective mechanism of action and resistance and the reference.

> We have now included such a table.

Images quality and size should be improved.

> We have now included high resolution images.

In the text some typos are present and English should be revised (e.g. in Figure 3A the authors use the term “dimerisation” instead of “dimerization” present in the whole text).

>Most of the typos resulted from a mixed use of US and Irish English spelling. We have corrected this and now use US spelling throughout.

The authors should render the aim of the review more appealing to the readers. For instance, by modifying the sentences “For the remainder of the review we focus on discussing less well reviewed areas of MAPK signaling” lane 47-50 or “Thus, rather than trying to exhaustively review conflicting details we focus on elaborating common themes through illustrative example” lane 152-153.

>We have edited these sentences.

Reviewer 2 Report

The review article "Targeting MAPK signaling in cancer: mechanisms of drug resistance and sensitivity" provides a high quality and comprehensive analysis of the role of MAPK pathways in the response of cancer cells to therapy. The main feature of the manuscript comparing to other recent reviews is the focus on the reaction of downstream MAPK pathways to stress conditions in cancer cells. The manuscript is presented by recognized authors in the field with impressive background and perfectly fits the subject of the special issue "Targeting MAPK in Cancer". It is well written, addresses all the questions raised and could be recommended for publication in the present form.

Author Response

Response to Reviewer 2 (our responses are marked by >)

The review article "Targeting MAPK signaling in cancer: mechanisms of drug resistance and sensitivity" provides a high quality and comprehensive analysis of the role of MAPK pathways in the response of cancer cells to therapy. The main feature of the manuscript comparing to other recent reviews is the focus on the reaction of downstream MAPK pathways to stress conditions in cancer cells. The manuscript is presented by recognized authors in the field with impressive background and perfectly fits the subject of the special issue "Targeting MAPK in Cancer". It is well written, addresses all the questions raised and could be recommended for publication in the present form.

> We are delighted that the reviewer finds this work publishable in the current form.